# Job Demands, Work Functioning and Mental Health in Dutch Nursing Home Staff during the COVID-19 Outbreak: A Cross-Sectional Multilevel Study

**DOI:** 10.3390/ijerph19074379

**Published:** 2022-04-06

**Authors:** Ylse van Dijk, Sarah I. M. Janus, Michiel R. de Boer, Wilco P. Achterberg, Corne A. M. Roelen, Sytse U. Zuidema

**Affiliations:** 1Department of General Practice and Elderly Care Medicine, University of Groningen, University Medical Center Groningen, 9700 AD Groningen, The Netherlands; s.i.m.janus@umcg.nl (S.I.M.J.); m.r.de.boer@umcg.nl (M.R.d.B.); s.u.zuidema@umcg.nl (S.U.Z.); 2Department of Public Health and Primary Care, Leiden University Medical Center, 2300 RC Leiden, The Netherlands; w.p.achterberg@lumc.nl; 3Department of Health Sciences, University of Groningen, University Medical Center Groningen, 9700 AD Groningen, The Netherlands; c.a.m.roelen@umcg.nl

**Keywords:** COVID-19, nursing homes, job demands, work functioning, depressive symptoms, burnout

## Abstract

COVID-19 posed enormous challenges for nursing home staff, which may have caused stress and mental health problems. This study aimed to measure the prevalence of mental health problems among nursing home staff and investigate the differences in job demands, work functioning and mental health between staff with and without COVID contact or COVID infection and across different levels of COVID worries. In this cross-sectional study, 1669 employees from 10 nursing home organizations filled in an online questionnaire between June and September 2020. The questionnaire measured the participants’ characteristics, COVID contact, infection and worries, job demands, work functioning, depressive symptoms and burnout. Differences were investigated with multilevel models to account for clustering at the organization level. Of the participants, 19.1% had high levels of depressive symptoms and 22.2% burnout. Job demands, work functioning, depressive symptoms and burnout differed between participants who never worried and participants who often or always worried about the COVID crisis. Differences were smaller for participants with and without COVID contact or infection. Most models improved when clustering was accounted for. Nursing homes should be aware of the impact of COVID worries on job demands, work functioning and mental health, both at the individual and organizational level.

## 1. Introduction

Nursing home staff are facing enormous challenges during the COVID-19 pandemic. These challenges include a high emotional burden, the fear of spreading the virus and a high workload. The frailty of nursing home clients and their increased risks of fatal consequences from the virus make the work for the staff emotionally demanding [1,2,3]. It also causes fear in staff of getting infected and subsequently spreading COVID-19 among clients [1,3]. The shortage of personal protective equipment, as experienced at the beginning of the pandemic, might have increased this fear [3,4,5]. The staff’s concern for the clients is increased by the strain on delivery of care and diminished routine due to (the constantly changing) restrictive measures [1,3]. Additionally, nursing home staff faces moral dilemmas while implementing measures such as the visit ban or quarantine [2,5]. Furthermore, they have to cope with staff shortages and a high workload as a result of colleagues in quarantine or in isolation due to a COVID infection [1,3,6]. Overall, these challenges are likely to cause stress and chronic mental health problems in healthcare staff. 

Studies focusing on healthcare staff during COVID-19 report that during or right after an outbreak, healthcare staff experiences stress. The experienced stress is often associated with the fear of getting infected, concern for family and changes in work environment or workload [7,8,9]. Especially healthcare staff working with infected patients are at a high risk for stress during a pandemic [8,10]. COVID-19 literature about healthcare staff reports prevalence rates for several mental health problems such as anxiety, depression, burnout and post-traumatic stress disorders. As reported in recent systematic reviews and meta-analyses, the pooled prevalence rates for burnout and depressive symptoms in healthcare staff during COVID-19 were, respectively, 28% and between 20.2% and 24.3% [11]. 

In many countries, nursing home clients made up substantial proportions of COVID-19-related deaths [12]. In the Netherlands, a total of 8.482 nursing home clients were registered with a COVID-19 infection at the end of June 2020. This is approximately 7% of the Dutch nursing home population [13], but the real number of infected clients is expected to be much higher. Dutch nursing homes provide multidisciplinary care for the elderly and chronically ill, comprising a wide range of inpatient and outpatient medical and social care. Furthermore, many nursing home organizations provide geriatric rehabilitation and homecare [14,15]. The care is provided by the multidisciplinary teams employed by the nursing homes. These teams include registered nurses, nurse assistants, nursing home physicians and other healthcare professionals (e.g., physiotherapists, psychologists, occupational therapists) [14,16].

Limited studies focus on nursing home staff during COVID-19. Some of these studies describe the experiences of nursing home staff during COVID-19 and confirm the fear of getting infected, concern for family and changes in workload as sources of stress [1,3,6,14,17]. The implementation of guidelines and the emotional burden were experienced as additional sources of stress. Other studies report prevalence rates of several mental health outcomes in nursing home staff, such as anxiety, post-traumatic stress disorders, mood disturbance, satisfaction and moral injury [18,19,20]. The prevalence of burnout and depression is especially interesting in nursing home staff. Their work is considered emotionally demanding and stressful, because they are repeatedly confronted with people’s needs, problems and suffering [21,22,23,24]. Therefore, nursing home employees are susceptible to developing burnout and depression. It is expected that these mental health problems eventually influence the quality of care through diminished work functioning [25]. Studies reporting prevalence rates in healthcare staff or nursing home staff often overlook the underlying job demands that cause stress. Potential job demands related to mental health problems are quantitative, cognitive and emotional job demands [26,27,28]. 

The aim of the present study is to gain insight into the prevalence of mental health problems and investigate the differences in job demands, work functioning and mental health between nursing home staff with and without COVID contact or COVID infection and across different levels of COVID worries. We expect that the first period (February–June 2020) of the COVID-19 pandemic in the Netherlands has a large impact on nursing home staff, because of the sudden measures taken by the government and organizations to restrict the spread of a virus largely unknown at that moment. These unexpected changes in the working environment are likely to increase stress in nursing home staff. Investigating the differences in job demands, work functioning and mental health between nursing home staff with and without COVID contact or COVID infection and across different levels of COVID worries is important for the implementation of psychosocial support and preventive measures in nursing home organizations during ongoing waves of mutant SARS-CoV-2 viruses.

## 2. Materials and Methods

### 2.1. Design and Setting 

In this cross-sectional study, data were collected from 25 June 2020 to 3 September 2020, which is after the first wave of COVID-19 infections in The Netherlands. 

### 2.2. Participants 

The directors of nursing home organizations, who were participating in the MINUTES study [15], were asked by e-mail in May 2020 to participate in the current study. In addition, the directors of organizations from a nursing home research network of Northern Netherlands (UNO-UMCG) were asked to participate. In total, 40 nursing home organizations were contacted of which 10 agreed to participate in the study. 

Data were collected through an online questionnaire in Redcap. The link to the questionnaire was sent to the human resource managers of the 10 participating nursing home organizations. The human resource managers distributed the questionnaire to their organizations’ staff. Questionnaire responses were sent to the researchers; neither the human resource managers nor the nursing home organizations had insight into questionnaire responses on the individual level.

### 2.3. Ethics 

The Medical Ethical Review Board of the University Medical Centre Groningen approved of the study. Informed consent was given by the participants in Redcap by clicking on the consent button. If no consent was provided, the questionnaire could not be filled in. 

### 2.4. Data Collection 

The first part of the questionnaire included questions about the participants’ characteristics and the COVID-19 circumstances at the workplace. The second part of the questionnaire was developed based on validated scales to measure (1) job demands, (2) work functioning and (3) mental health.

### 2.5. Participant Characteristics

Participant characteristics included age, gender, organization and function type (residential care employee, homecare employee, healthcare professionals (e.g., physiotherapist, physician), and non-care employee (e.g., cleaning, kitchen, and office staff)). To describe the COVID-19 circumstances at the workplace, they were asked (1) whether they had been in direct contact with COVID-infected clients (yes/no), (2) whether they got infected with COVID-19 themselves (yes/no) and (3) whether they worried about the COVID-19 crisis (“I worry about the COVID-19 crisis”; 0: never; 1: rarely; 2: sometimes; 3: often; 4: always). Similar statements to measure COVID worries were used in a previous study [29].

### 2.6. Job Demands 

Job demands were measured using the Copenhagen Psychosocial Questionnaire (COPSOQ) [30]. Three COPSOQ scales were used: quantitative demands (four items, Cronbach’s α = 0.84), cognitive demands (four items, α = 0.82) and emotional demands (three items, α = 0.73). All items have to be answered on a five-point scale from 0 (“always”) to 5 (“never”). The answers were converted to a score of 0–100, whereby higher scores express higher levels of the job demand. The internal consistencies of the COPSOQ scales were similar to those previously reported (α = 0.65, α = 0.78, and α = 0.87, respectively) [30]. 

### 2.7. Work Functioning

Work functioning is the ability to meet work demands given the state of (physical and mental) health [31]. Work functioning was measured using the 10-item Work Role Functioning Questionnaire [32]. The WRFQ assesses work scheduling, output demands, mental demands, physical demands and flexibility demands. All items were answered on a four-point scale from (0 = difficult all the time/most of the time to 3 = difficult none of the time). It was also possible to answer “not applicable to my work”; this response was treated as a missing value. The total score was calculated by averaging the score on the ten items and multiplying the average score by 25 to obtain a score between 0 and 100. Higher scores indicate better work functioning. The internal consistency found in the current study (α = 0.92) is similar to that previously reported (α = 0.96) [31]. 

### 2.8. Mental Health

Mental health comprised two aspects: depressive symptoms and burnout. Depressive symptoms were measured using the depression component of the Hospital Anxiety and Depression Scale (HADS) [33]. The depression component of the HADS consists of seven items with different four-point scales (0–3 points), for example, “not at all” to “most of the time”. A total score was calculated by adding the scores on the seven items. The internal consistency of the scale (α = 0.85) was similar to that previously reported (α = 0.67–0.90) [34]. A total score of ≥8 indicates the presence of depressive symptoms; this cut-off was also used for the calculation of the prevalence rate [35]. Burnout was measured using the exhaustion component of the Burnout Assessment Tool (BAT) [36]. In the BAT, the exhaustion component consists of eight items, which were answered on a five-point scale from 1 = never to 5 = always. A total score was calculated by adding the scores on the items and dividing by the total number of items. The internal consistency found in this study (α = 0.93) is similar to that previously reported (α = 0.94) [37]. The cut-off score of 3.00 for exhaustion in the Dutch working population was used to categorize the participants as having high (>3.00) or low (≤3.00) risk of burnout [36].

### 2.9. Analysis

The Statistical Package for the Social Sciences (SPSS version, v. 26.0, IBM, Armonk, NY, USA) was used for all analyses. Descriptive statistics were used to depict characteristics. Frequencies were used for categorical variables, and means and standard deviations for continuous variables. Prevalence rates of depressive symptoms and burnout were calculated by using the cut-off values. 

The quantitative demands (QD) experienced by participants who had been in contact with infected clients were compared to the QD experienced by participants without COVID contact using a linear multilevel model to account for clustering at the organization level. In the models, employees (level 1) were nested in nursing home organizations (level 2). First, an intercept-only model was built without adding random coefficients to the model. Next, COVID contact was added as a fixed effect. Random intercepts, slopes and the covariance between intercepts and slope were added to the models when the model fit (based on Likelihood Ratio tests) improved. Effect estimates were expressed as beta (β) with 95% confidence intervals (CI). Separate linear multilevel models were used for cognitive demands (CD), emotional demands (ED), work functioning (WF), burnout (BO), and depressive symptoms (DS). 

Likewise, the QD, CD, ED, WF, BO and DS of participants who had been infected with COVID were compared to participants without COVID infection. Finally, QD, CD, ED, WF, BO and DS were compared across different levels of COVID worries: always, often, sometimes, rarely, and never.

Sensitivity analyses were performed for care staff only (homecare employees and residential care employees). The results of this analysis are reported in the Appendix A section.

Considering our focus on fixed regression parameters, the maximum likelihood (ML) method was used for estimating parameters in the linear multilevel models [38]. The residual method was used as the degrees of freedom method because our sample size is sufficiently large and because in most of the models the Variance Components type of covariance was used [39].

## 3. Results

### 3.1. Descriptive Characteristics of the Participants

The questionnaire was filled in by 2114 employees from 10 nursing home organizations. A total of 445 participants were excluded from the analysis because information regarding age, organization or function was missing. Consequently, the data of 1669 questionnaires were eligible for analysis (see Table 1). About one in five participants suffered from burnout or depressive symptoms (respectively, 22.2% and 19.1%). 

### 3.2. Differences in Job Demands, Work Functioning and Mental Health between Staff with and without COVID Contact or COVID Infection and Across Different Levels of COVID Worries 

The multilevel analysis showed that some models improved when random intercepts and slopes were included (see Table 2 and Table 3). 

In general, our results indicated small differences in job demands, work functioning and mental health between participants who had been in contact with COVID-infected clients or had been infected with COVID and those who were not (see Table 2 and Table 3).

Compared to the reference group of participants who always worried about the COVID-19 crisis, the participants who never worried experienced lower quantitative demands (β −22.76 [−31.24 to −14.28]), cognitive demands (β −14.72 [−22.69 to −6.75]), emotional demands (β −18.03 [−26.21 to −9.84], better work functioning (β 11.65 [2.10–21.21]) and fewer depressive (β −4.82 [−6.65 to −2.99]) and burnout symptoms (β −0.71 [1.06–0.36]) (see Table 2 and Table 3). There were also substantial differences between staff who never worried and those who often worried about the COVID-19 crisis. Differences between participants who never worried and those who sometimes or rarely worried varied in magnitude and direction between the outcomes. For example, those who rarely worried had a better score on cognitive demands but a poorer score on quantitative demands than those who never worried (see Table 2 and Table 3). So, within some of the levels of COVID worries (never, rarely, sometimes), participants who worried less did not always score better on job demands, work functioning and mental health than participants who worried more often.

The outcomes of the sensitivity analyses (see Appendix A) showed similar results for care staff only (homecare employees and residential care employees).

## 4. Discussion

This cross-sectional study used a multilevel analysis to gain insight into the prevalence of mental health problems in nursing home staff and investigate the differences in job demands, work functioning and mental health between nursing home staff with and without COVID contact or COVID infection and across different levels of COVID worries. Mental health problems were common among nursing home staff, with about one in five suffering from burnout or depressive symptoms. Small differences were found in job demands, work functioning and mental health between participants who had been in contact with COVID-infected clients or had been infected with COVID-19 and those who were not. Large differences were found in job demands, work functioning and mental health between participants who never worried and participants who often or always worried.

In the current study, the prevalence rates of burnout and depressive symptoms were, respectively, 22.2% and 19.1%. Comparison with reported prevalence rates in healthcare staff (28% for burnout and between 20.2% and 24.3% for depressive symptoms [11]) is difficult for these components of our questionnaire, since different scales were used to report depressive symptoms (i.e., GAD-7, DAS-21) and burnout (i.e., MBI reporting the prevalence of the three different components of burnout). Besides, most of the included studies concern Chinese healthcare staff [11]. Cultural aspects might contribute to different results, and possibly, prevalence rates were dependent on the timing of the studies. According to the literature about earlier pandemics [40,41], it is expected that only acute stress reactions are present in this phase of the pandemic. Longitudinal studies are needed to study the long-term effects such as depressive symptoms and burnout in more depth.

Only a few studies on pandemics focus on the level of worries. In the current study, relatively high levels of COVID worries were found, given that almost half of the participants often or always worried about COVID-19. An Italian study reported lower levels of COVID worries among healthcare staff compared to the general population. The authors attributed these lower levels of COVID worries in healthcare staff to the fact that they had professional and academic knowledge of COVID-19 [42]. The high levels of worries in the current study might be explained by the difficult work situation in nursing homes. The staff works with highly vulnerable clients and is worried about infecting the clients and infecting their family members [1,3,6].

The current study showed large differences in job demands, work functioning and mental health between participants who never worried and participants who often or always worried. The differences in job demands were considered large based on the minimally important score difference (of 10.8) that was reported for this scale in a previous study [43]. Whereas only a limited number of studies on pandemics have focused on worries, the fear of getting infected during pandemics has been described in many studies [1,3,6,8,14,44]. Some of these studies reported associations between this fear and symptoms of anxiety or depression [45,46]. The authors of a previous study described worrying as a form of mental coping with emotional stress, by thinking of possible solutions for threatening events [47]. However, worrying also results in doubts about the problem-solving ability and being pessimistic about problem-solving outcomes [47]. Staff might therefore perceive their coping efforts as less effective and thus experience more negative moods and depressive symptoms. This might also apply to the current study in which the staff who often or always worried experienced more depressive symptoms than those who never worried. 

In the current study, the differences between the staff who had been in contact with infected clients and those who had not been in contact are small and do not seem clinically relevant. In line with these findings, a study on Dutch healthcare staff during COVID-19 showed differences in the levels of physical exhaustion [48] and a study on Spanish nursing home staff showed differences in the levels of emotional exhaustion [20] between the staff who had been in contact with infected clients and those who had not been in contact. Given the small effect sizes in these studies, the clinical relevance of these differences is speculative. 

Furthermore, small differences were found in depressive symptoms and burnout between staff who had been infected with COVID-19 and those who were not. This was in line with a previous study among healthcare staff [49]. The differences in depressive symptoms between those with and without a COVID infection could be attributed to social isolation during quarantine as described in that study [49]. However, interviews with nursing home staff in the Netherlands showed that all nursing home employees (infected or not) had fewer social contacts than others in society due to their awareness of the possibility of bringing the virus into the nursing home [14]. This could be the reason for finding small differences in depressive symptoms between staff who had been infected and staff who were not. Only a few studies reported associations between COVID-19 infections and burnout in healthcare staff [49,50]. It would be interesting to focus on the effect of COVID infections on (mental) health-related factors in future studies, especially in combination with upcoming literature about long-COVID symptoms [51].

The multilevel analysis showed an improvement of some models when random intercepts and slopes on the level of the organization were included. Clustering at the organization level might suggest that organizational factors influence the outcomes of nursing home staff. One could think of the style and quality of leadership, the different COVID measures taken by organizations and the psychosocial support that was arranged during the COVID-19 outbreak. A recent literature review reports a lack of studies investigating the effectiveness of interventions to improve the mental health of frontline staff during pandemics [7]. However, it is recognized that interventions should be properly planned and include both organization-directed and personal-directed aspects [7,52]. At the organizational level, it is important to facilitate clear and updated communication, provide flexible schedules and organize team meetings to address conflicts and strengthen teamwork [7]. Person-directed measures should enable employees to manage their experiences and increased job demands by interventions promoting a healthy lifestyle and self-care or by providing psychosocial support [7]. Providing psychosocial support using cognitive-behavioral techniques could be used to help staff to stop negative cycles of thoughts and to change the way they respond to distress [7,53].

To our knowledge, this is the first study that uses a multilevel analysis and focuses on work functioning during COVID-19. We used existing scales, which makes future comparisons between studies possible. This study also has some limitations. First, the use of a self-administered online questionnaire could have led to bias such as over reporting of problems. Second, the response rate within the nursing home organizations was low (9%). Although there was a large number of participants from different organizations distributed through the Netherlands, our descriptive statistics may not be representative of the overall nursing home staff. Third, the number of participants differed across organizations. For example, the descriptive statistics may be biased because a relatively large percentage (*n* = 397, 24%) of participants originated from one organization. With regard to the other results, this limitation is covered because associations were investigated and a multilevel analysis was used. Fourth, we computed a large number of associations. Some of our findings may therefore be so-called chance findings. Fifth, future comparisons with our scores of work functioning have to take into account that we coded the answers on the questionnaire differently. We used four answer options instead of five, merging “often” and “always” together. Lastly, as the present study had a cross-sectional design, we cannot infer causal relations between COVID-19 and possible work- or (mental) health-related problems. Further studies should focus on the development of work functioning and mental health problems during the COVID-19 pandemic using a longitudinal cohort approach.

## 5. Conclusions

The results of the study showed that mental health problems are common among nursing home staff. Large differences were found in job demands, work functioning, depressive symptoms and burnout between those who often or always worry about the COVID-19 crisis and those who never worry about the COVID-19 crisis. Nursing homes should be aware of the impact of COVID worries on job demands, work functioning and mental health, both at the individual and organizational level. 

## Figures and Tables

**Table 1 ijerph-19-04379-t001:** Descriptive characteristics of the participants (*n* = 1669 nursing home employees).

Variables	
Age, mean (SD, range)	45.7 (12.8, 17–68)
Gender ^1^, *n* (%):	
Men	139 (8.4)
Women	1525 (91.6)
Function, *n* (%):	
Homecare employee	85 (5.1)
Residential care employee	1063 (64.7)
Healthcare professional	176 (10.5)
Non-care employee	345 (20.7)
COVID circumstances, *n* (%):	
COVID contact ^2^ (yes)	482 (30.0)
COVID infection ^3^ (yes)	102 (6.4)
COVID worries ^4^	
(0) Never	20 (1.2)
(1) Rarely	116 (7.2)
(2) Sometimes	765 (47.4)
(3) Often	630 (39.0)
(4) Rarely	83 (5.1)
Job demands:	
Quantitative demands ^5^, mean (SD, range)	55.3 (18.1, 0–100)
Cognitive demands ^6^, mean (SD, range)	63.9 (16.60, 0–100)
Emotional demands ^7^, mean (SD, range)	61.5 (17.7, 0–100)
Work functioning ^8^, mean (SD, range)	78.9 (16.7, 0–100)
Burnout ^9^, mean (SD, range)	2.4 (0.7, 1–5)
- Prevalence, *n* (%)	330 (22.2)
Depressive symptoms ^10^ (SD, range)	4.1 (3.7, 0–21)
- Prevalence, *n* (%)	258 (19.1)

Missing (*n*, %): ^1^ (5, 0.3), ^2^ (62, 3.7), ^3^ (66, 4.0), ^4^ (55, 3.3), ^5^ (92, 5.5), ^6^ (96, 5.8), ^7^ (94, 5.6), ^8^ (430, 25.8), ^9^ (183, 11.0), ^10^ (319, 19.1).

**Table 2 ijerph-19-04379-t002:** Differences between nursing home staff with and without COVID contact or COVID infection and across different levels of COVID worries in job demands. Reference group: COVID contact (no), COVID infection (no), COVID worries (always).

	Job Demands								
	Quantitative Demands			Cognitive Demands			Emotional Demands		
	B	*p*	95% CI	B	*p*	95% CI	B	*p*	95% CI
Intercept	55.52 ^1^		52.96 to 58.08	63.89		63.07 to 64.71	61.75 ^1^		59.62 to 63.87
COVID contact (no)	0			0			0		
COVID contact (yes)	2.72 ^1^	0.006	0.76 to 4.67	1.94	0.033	0.15 to 3.72	5.20 ^1^	<0.001	3.30 to 7.11
COVID infection (no)	0			0			0		
COVID infection (yes)	0.48 ^2^	0.870	−5.93 to 6.89	−0.16	0.928	−3.55 to 3.24	4.91 ^1^	0.007	1.34 to 8.48
COVID worries									
always (4)	0			0			0		
often (3)	−8.13 ^1^	<0.001	−12.14 to −4.11	−10.34	<0.001	−14.16 to −6.52	−12.55 ^1^	<0.001	−16.43 to −8.67
sometimes (2)	−14.17 ^1^	<0.001	−18.16 to −10.19	−14.64	<0.001	−18.43 to −10.86	−20.07 ^1^	<0.001	−23.91 to −16.22
rarely (1)	−17.69 ^1^	<0.001	−22.65 to −12.73	−16.91	<0.001	−21.60 to −12.22	−25.13 ^1^	<0.001	−29.92 to −20.35
never (0)	−22.76 ^1^	<0.001	−31.24 to −14.28	−14.72	<0.001	−22.69 to −6.75	−18.03 ^1^	<0.001	−26.21 to −9.84

^1^ included random intercepts; ^2^ included random intercepts and slopes; % missing (range): QD: 6–7%, CD: 6–7%, ED: 6–7%.

**Table 3 ijerph-19-04379-t003:** Differences between nursing home staff with and without COVID contact or COVID infection and across different levels of COVID worries in work functioning and mental health. Reference group: COVID contact (no), COVID infection (no), COVID worries (always).

	Work Functioning and Mental Health								
	Work Functioning			Depressive Symptoms			Burnout		
	B	*p*	95% CI	B	*p*	95% CI	B	*p*	95% CI
Intercept	78.89		77.96 to 79.82	4.09 ^1^		3.58 to 4.60	2.41 ^1^		2.29 to 2.52
COVID contact (no)	0			0			0		
COVID contact (yes)	−1.53	0.132	−3.53 to 0.46	0.36 ^1^	0.111	−0.08 to 0.79	0.05 ^1^	0.229	−0.03 to 0.13
COVID infection (no)	0			0			0		
COVID infection (yes)	−5.66	0.004	−9.49 to −1.83	1.82 ^1^	<0.001	1.03 to 2.61	0.33 ^1^	<0.001	0.17 to 0.48
COVID worries									
always (4)	0				0			0	
often (3)	8.23	<0.001	3.72 to 12.74	−2.93 ^1^	<0.001	−3.83 to −2.03	−0.46 ^1^	<0.001	−0.64 to −0.29
sometimes (2)	13.17	<0.001	8.70 to 17.65	−4.58 ^1^	<0.001	−5.48 to −3.69	−0.76 ^1^	<0.001	−0.93 to −0.59
rarely (1)	15.05	<0.001	9.61 to 20.50	−4.52 ^1^	<0.001	−5.62 to −3.41	−0.84 ^1^	<0.001	−1.05 to −0.63
never (0)	11.65	0.017	2.10 to 21.21	−4.82 ^1^	<0.001	−6.65 to −2.99	−0.71 ^1^	<0.001	−1.06 to −0.36

^1^ Included random intercepts; % missing: WF: 26–27%, DS: 19–20%, BO: 11–12%.

## Data Availability

The data presented in this study are available on request from the corresponding author. After completing the project, the data will be made publicly available.

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
