# Peer review of "Job Demands, Work Functioning and Mental Health in Dutch Nursing Home Staff during the COVID-19 Outbreak: A Cross-Sectional Multilevel Study"

_ijerph, 2022, doi:10.3390/ijerph19074379_

Round 1

Reviewer 1 Report

Overall, I think it is a very interesting and topical article, where the authors analyse the problem in a rigorous way. As for the introduction, I think it is appropriate that reference is made to the JDR model, where demands and resources are predictors of health problems and burnout, in particular. 
However, in the treatment of the data and in the objectives it sets out, it seems to forget the theoretical model from which it starts; in fact, I find it difficult to understand why demands and work functioning are considered as dependent variables, since according to the proposed model, they should be considered as independent variables, and perhaps incorporated into the models in order to know their value and impact on the measurement of health and burnout. 
Finally, it should be noted that the JDR model does consider resources, and the research makes no mention of any variable that measures them. Therefore, it is necessary to reconsider whether this model is the one to follow in the research.

Reviewer 2 Report

Dear Authors,

I'm grateful to have the chance of reviewing your interesting work about mental health in worker of nursing home during COVID-19 pandemic. It is a topic of great importance, becuase these healthcare workers  have cared for the weakest and more vulnerable patients. 

I found that your work has great potentiality. Below you can find my comments.

Introduction:

-you mention previous studies regarding nursing home workers in other pandemics, comparing them to hospital workers. The comparison sounds good, but why don't you include the current literature after over two years of COVID-19 pandemic?

-you should add a brief definition of work functioning since it is a variable you aare using. 

-I don't understand the variable "worrying about COVID": what does it mean? Could you please specify? Worrying about what? infection, increasing job demand, radical changes in life and work....you mention something in lines 268 and following but you need to explain better. 

Method:

-you should put the paragraph "setting" in the Introduction

Discussion:

-you should add as a limitation that the questionnaire was self administered online, this is another bias in the results

-you should add practical implications that can derive from your study; you left the reader with a "so what...?" feeling. 

-line 277 is repeated in line 286. 

-in the Results paragraph you state: "those who rarely worried had a better score on cognitive demands but a poorer score on quantitative demands than those who never worried". Could you explain? 

-I suggest to add some updated literature references, that can help you explain some topics you address in the paper and provide practical implications:

https://doi.org/10.1111/jocn.14165

https://pubmed.ncbi.nlm.nih.gov/34574378/ 

https://pubmed.ncbi.nlm.nih.gov/26084675/ 

https://www.mdpi.com/2071-1050/13/24/13869 

https://www.cochranelibrary.com/cdsr/doi/10.1002/14651858.CD013779/full 

https://pubmed.ncbi.nlm.nih.gov/34729818/ 

Good work and best wishes

Round 2

Reviewer 1 Report

The authors' comments and suggestions do not solve the problems of the theoretical fit of the study. Although, at least it avoids theoretical confusion

Reviewer 2 Report

Dear Authors,

I think you properly addressed my comments and suggestions and improved your manuscript. 

Best regards